# Prognosis of Acute Ischaemic Stroke Patients with Cancer: A National Inpatient Sample Study

**DOI:** 10.3390/cancers13092193

**Published:** 2021-05-03

**Authors:** Tiberiu A. Pana, Mohamed O. Mohamed, Mamas A. Mamas, Phyo K. Myint

**Affiliations:** 1Keele Cardiovascular Research Group, Centre for Prognosis Research, Keele University, Stoke-on-Trent ST4 6QG, UK; M.mohamed@keele.ac.uk (M.O.M.); m.mamas@keele.ac.uk (M.A.M.); phyo.myint@abdn.ac.uk (P.K.M.); 2Institute of Applied Health Sciences, School of Medicine, Medical Sciences & Nutrition, University of Aberdeen, Aberdeen AB25 2ZD, UK

**Keywords:** ischaemic stroke, cancer, mortality, revascularisation, thrombolysis, thrombectomy

## Abstract

**Simple Summary:**

The impact of cancer on the acute prognosis of stroke patients remains largely unknown. Furthermore, the usage of interventions aiming to restore cerebral blood flow in ischaemic stroke, such as thrombolysis and thrombectomy, remains uncharacterised in cancer patients. We aimed to delineate these relationships using a sample representative of 1,106,045 acute ischaemic stroke admissions across the US between 2015–2017, 3.51% of whom had cancer. We found that non-metastatic and metastatic cancers were associated with significantly increased odds of in-hospital mortality, prolonged hospitalisation and decreased odds of home discharge. We also determined that both thrombolysis and thrombectomy offset the association between non-metastatic cancer and in-hospital mortality. Thrombectomy offset the association between metastatic cancer and in-hospital mortality. We conclude that cancer patients warrant robust stroke prevention, given their increased odds of adverse outcomes. Thrombolysis and thrombectomy should be considered routinely in stroke patients with cancer unless otherwise contraindicated.

**Abstract:**

Whilst cancer is a risk factor for acute ischaemic stroke (AIS), its impact on AIS prognosis between metastatic and non-metastatic (MC and NMC) disease is poorly understood. Furthermore, the receipt of intravenous thrombolysis (IVT) and endovascular thrombectomy (ET) and their outcomes is poorly researched. AIS admissions from the National Inpatient Sample (NIS) were included (October 2015–December 2017). Multivariable logistic regressions adjusting for a wide range of confounders analysed the relationship between NMC and MC and AIS in-hospital outcomes (mortality, prolonged hospitalisation >4 days and routine home discharge). Interaction terms with IVT and ET were also computed to explore their impact amongst cancer patients. A total of 221,249 records representative of 1,106,045 admissions were included. There were 38,855 (3.51%) AIS admissions with co-morbid cancer: NMC = 53.78% and MC = 46.22%. NMC was associated with 23% increased odds of in-hospital mortality (odds ratio (95% confidence interval) = 1.23 (1.07–1.42)), which was mainly driven by pancreatic and respiratory cancers. This association was entirely offset by both IVT and ET. MC was associated with two-fold increased odds of in-hospital mortality (2.16 (1.90–2.45)), which was mainly driven by respiratory, pancreatic and colorectal cancers. This association was only offset by ET. Both NMC and MC were significantly associated with prolonged hospitalisation and decreased odds of routine discharge. Cancer patients are at higher odds of acute adverse outcomes after AIS and warrant robust primary prevention. IVT and ET improve these outcomes and should thus be offered routinely unless otherwise contraindicated in this group of stroke patients.

## 1. Introduction

Malignancy is associated with increased risk of acute ischaemic stroke (AIS) [1]. This is mediated through a variety of mechanisms, including hypercoagulability [2] and shared risk factors [1]. Cancer is therefore important to consider in AIS, as it may not only be more prevalent amongst AIS patients [3], but may also be associated with adverse outcomes [4,5]. Despite that fact that cancer is a heterogenous disease, the distribution of organ-specific primary cancer types amongst a representative sample of AIS patients remains undescribed. Furthermore, the individual associations between each cancer type and adverse AIS outcomes remain largely unknown.

Revascularisation therapies (intravenous thrombolysis (IVT) and endovascular thrombectomy (ET)) significantly improve AIS outcomes [6,7] and are thus recommended routinely in eligible patients [8]. Nevertheless, co-morbid cancer may hinder their use given that this population is more likely to exhibit contraindications to IVT or ET [1]. Current guidelines recommend IVT in AIS patients with systemic malignancy provided they have a life expectancy >6 months and no contraindications [8]. These recommendations are based on limited evidence, as landmark randomised controlled trials studying IVT [9,10,11,12] or ET [7] have excluded cancer patients. Previous smaller scale observational studies of AIS patients with cancer have found no increased haemorrhagic complications or mortality associated with IVT [13,14,15]. Nevertheless, guidelines currently provide no specific recommendations regarding ET for this population [8], with the efficacy and safety of ET amongst AIS patients with cancer being largely unclear. While several small retrospective studies have found no association between cancer and adverse outcomes in AIS patients undergoing ET [16,17], others have identified significantly higher mortality in cancer patients [18].

This drives the need for a comprehensive description of the association between cancer and AIS outcomes in contemporary clinical practice and whether revascularisation therapies have an effect on these outcomes. In this study, using a representative sample of AIS admissions in the United States (US) between 2015–2017, we sought examine the prevalence of comorbid cancer among patients with AIS and its association with in-hospital outcomes, also stratifying by metastatic disease. We also aimed to examine the effect of IVT/ET on outcomes in cancer patients through the use of interaction terms.

## 2. Materials and Methods

This study was conducted in accordance with the principles of the Declaration of Helsinki (1975) and later amendments. The data that support the findings of this study are available from the corresponding author upon reasonable request.

### 2.1. Data Source and Inclusion Criteria

The National Inpatient Sample (NIS) is a publicly available database containing >7 million annual hospital admission records. NIS contains admission records representing a 20% stratified sample of all community hospital admissions in the United States. Using the provided sampling weights, the NIS data can be used to provide national estimates for the sampling population, representative of ~95% of the US population [19,20]. Prior to undertaking this project, all authors completed the Healthcare Cost and Utilisation Project (HCUP) Data Use Agreement Training Tool. All authors also read and signed the Data Use Agreement for Nationwide Databases. As the NIS is publicly available and contains no patient identifiable information, no ethical approval was needed. Using data files containing annual admissions between 2015–2017, all records with a primary diagnosis of ischaemic stroke (International Classification of Disease—Tenth Edition (ICD-10) codes I63.0–I63.9) were extracted. Only cases admitted between October 2015–December 2017 were included due to a change in co-morbidity coding (ICD-9 to ICD-10) occurring after September 2015 [20]. Elective admissions and those with missing data on key variables were excluded.

### 2.2. Statistical Analysis

All analyses were performed using Stata 15.1SE, Stata Statistical Software. A 5% threshold of statistical significance was utilised for all analyses (*p* < 0.05). Analyses were performed following HCUP guidelines [21], utilising the provided discharge weights as probability weights and survey data analysis techniques stratifying by NIS stratum and year of admission [22] in order to account for patient clustering within hospitals and produce US-wide estimates [23].

#### 2.2.1. Outcomes

The following outcomes were analysed: (1) in-hospital mortality, (2) prolonged hospital stay in excess of 4 days and (3) routine discharge from hospital. Vital status upon hospital discharge (dead/alive) and the length of stay (LoS) in hospital are provided as standard variables in the NIS [24,25]. Prolonged hospitalisation was defined as LoS > 4 days, according to expert clinical opinion and previous studies assessing ischaemic stroke outcomes amongst patients admitted to hospital in the United States [26]. A dichotomous variable indicating patients hospitalised for >4 days was subsequently used as an outcome for LoS analyses. Discharge status was coded using the provided discharge destination [27]. All records of patients who were discharged against medical advice and those discharged to an unknown destination were excluded from the analyses prior to weighting (*n* = 2187 (0.99%)), allowing estimates for this particular outcome to be provided for 1,095,110 (99.01%) of AIS patients. Discharge destination was then dichotomised into “routine discharges” and other discharges (“home health care”, “short-term hospital”, “other facilities including intermediate care and skilled nursing home” and “died in hospital”). The “other discharges” category was subsequently used as a reference category in all analyses evaluating discharge destination.

#### 2.2.2. Exposures and Confounders

Co-morbid solid organ cancers (non-metastatic and metastatic) as well as the organ-specific types were the exposures of interest. Haematological cancers were not considered part of the exposure, as they form a distinct clinical entity with a different natural course. Nevertheless, all analyses were adjusted for co-morbid haematological malignancies. Co-morbid cancer diagnoses were identified using the Clinical Classification Software Refined (CCSR) codes (Appendix A) [28] and represent diagnoses assigned before or during the index acute ischaemic stroke hospitalisation. Patients with a previous history of cancer (ICD-10 Z85.x and Z86.0x) but without an ICD-10 code indicating an active cancer diagnosis (ICD-10 C00-C79) were included in the no active cancer group, the reference category. The Elixhauser co-morbidities solid tumour without metastases and metastatic cancer were used to ascertain the presence of metastatic disease [29]. Specific cancer types were also identified using the individual Clinical Classification Software Refined (CCSR) codes listed in Appendix A.

All models were adjusted for the following confounders: age, sex, ethnicity, Elixhauser co-morbidities (congestive heart failure, valvular disease, pulmonary circulatory disease, peripheral vascular disease, paralysis, other neurological disorders, chronic pulmonary disease, diabetes, hypothyroidism, renal failure, liver disease, peptic ulcer disease, acquired immune deficiency syndrome, rheumatoid arthritis, coagulopathy, obesity, weight loss, fluid and electrolyte disorders, anaemia, alcohol abuse, drug abuse, psychosis, depression and hypertension), previous history of cancer, haematological malignancies, other co-morbidities (dyslipidaemia, smoking, Parkinson disease, coronary heart disease, all-cause bleeding, pulmonary embolism, deep venous thrombosis, atrial fibrillation, arrhythmias other than atrial fibrillation, pneumonia (including aspiration), shock, previous cerebrovascular disease), hospital bedsize, location and teaching status and revascularisation therapy (thrombolysis, thrombectomy). Adjusting co-variates were selected based on clinical judgement and previous literature [5,14,15,30]. Previous history of cancer was identified using ICD10 codes Z85.x and Z86.0x. Elixhauser co-morbidities were determined using the HCUP Elixhauser co-morbidity software version 2020.1 [29]. Co-morbid conditions other than the Elixhauser co-morbidities were identified using ICD-10 codes (Appendix A).

#### 2.2.3. Descriptive Statistics

Characteristics were compared between AIS hospitalisations without cancer, those with non-metastatic cancer and those with metastatic disease. One-way analysis of variance and Pearson’s χ [2] test were employed to compare characteristics for continuous and categorical variables, respectively. The distribution of each primary cancer type amongst the included sample as well as the proportion of metastatic disease amongst each cancer type were determined. Characteristics were then also compared between AIS hospitalisations with the 5 most common types of primary cancers previously identified. No between-group tests of statistical significance were performed for these comparisons as the groups were not mutually exclusive.

#### 2.2.4. Association between Prevalent Cancer and Odds of Receiving Revascularisation Therapy

Multivariable logistic regressions were employed to analyse the relationship between co-morbid non-metastatic and metastatic cancer and the odds of receiving IVT and ET in hospital. All models were adjusted for the covariates listed above, with the exception of IVT or ET when this variable was used as the outcome.

#### 2.2.5. Association between Non-Metastatic and Metastatic Cancer and In-Hospital Outcomes

Multivariable logistic regressions were employed to analyse the relationship between co-morbid non-metastatic and metastatic cancer and in-hospital outcomes. Separate models containing interaction terms with IVT and ET were also computed to determine whether these relationships were modified by revascularisation therapies. All models were adjusted for the covariates listed above.

#### 2.2.6. Association between Non-Metastatic and Metastatic Cancer and In-Hospital Outcomes amongst Patients Undergoing Endovascular Thrombectomy

Further multivariable logistic regressions were performed in order to explore the relationship between co-morbid non-metastatic and metastatic cancer and in-hospital outcomes amongst AIS patients undergoing ET. Separate models including an interaction term with IVT were also computed in order to determine how outcomes were modified by whether these patients also received IVT pre-treatment. All models were adjusted for the covariates listed above.

#### 2.2.7. Association between the Five Most Common Primary Cancer Types and In-Hospital Outcomes

Multivariable logistic regressions were employed to analyse the relationship between the five most common primary cancer types and in-hospital outcomes, stratifying each cancer type by the presence of metastases. All five cancer types were simultaneously introduced in the same model. All models were adjusted for the covariates listed above as well as other co-morbid cancer types.

## 3. Results

Figure 1 details the study population. Out of 230,177 records extracted with a primary diagnosis of ischaemic stroke between October 2015–December 2017, a total of 8708 elective admission records as well as 220 records with missing data were excluded, yielding a total of 221,249 included records. After the application of sampling weights and the exclusion of strata with single sampling units, the included records were used to provide estimates for the population from which they were sampled: 1,106,045 hospitalisations with a primary diagnosis of AIS.

### 3.1. Descriptive Statistics

Figure 2 details the distribution of primary cancer types amongst the 38,855 AIS hospitalisations with co-morbid cancer, representing 3.51% of the entire included sample. The five most common types were: respiratory (9490 (24.42%)), prostate (4960 (12.77%)), breast (3375 (8.69%)), pancreatic (2640 (6.79%)) and colorectal cancers (2490 (6.41%)). There were 4750 (50.05%) hospitalisations with metastatic respiratory cancer, 1250 (25.20%) hospitalisations with metastatic prostate cancer, 1110 (32.89%) hospitalisations with metastatic breast cancer, 1875 (71.02%) hospitalisations with metastatic pancreatic cancer and 985 (39.56%) hospitalisations with metastatic colorectal cancer.

Table 1 and Appendix A detail the characteristics of the included population, representative of 1,106,045 AIS hospitalisations. The median (interquartile range) age was 72 (61–82) years and 50.41% were female. The median (interquartile range) length of stay was 3 (2–6) days. There were 20,895 (1.89%) hospitalisations with non-metastatic cancer and 17,960 (1.62%) with metastatic cancer. Median age ranged between 70 and 75 years, highest among those with non-metastatic cancer and lowest amongst those with metastatic cancer. The highest proportion of females was recorded amongst admissions with metastatic cancer (51.03%), followed by those without active cancer (50.47%) and those with non-metastatic cancer (46.88%). Compared to admissions without cancer, those with cancer had higher rates of prevalent chronic pulmonary disease, liver disease, coagulopathy and anaemia, but lower rates of congestive heart disease and diabetes. Compared to admissions without cancer, those with cancer had higher rates of in-hospital mortality, prolonged hospitalisation and lower rates of routine discharge.

Appendix A details the characteristics of the included hospitalisations with the five most common co-morbid cancer types. Those with prostate cancer were oldest, median (interquartile range)—79 (71–85) years, followed by those with breast cancer—74 (67–83) years, colorectal cancer—74 (65–82) years, respiratory cancers—71 (63–79) years and pancreatic cancer 71 (63–77) years. Hospitalisations with pancreatic cancer had the highest rate of in-hospital mortality (10.61%), followed by respiratory cancers (10.17%), colorectal cancer (6.22%), breast cancer (5.63%) and prostate cancer (3.73%).

### 3.2. Association between Prevalent Cancer and Odds of Receiving Revascularisation Therapy

Appendix A details the results of the multivariable logistic regressions evaluating the associations between non-metastatic and metastatic cancer and the odds of receiving thrombolysis or thrombectomy. Compared to hospitalisations without cancer, those with non-metastatic cancer had lower odds of receiving both IVT (odds ratio (95% confidence interval)—0.77 (0.66–0.90)) and ET (0.80 (0.63–0.9994)). Compared to hospitalisations without cancer, those with metastatic cancer had lower odds of receiving IVT (0.39 (0.32–0.47)), but not ET (0.87 (0.69–1.09)).

### 3.3. Association between Non-Metastatic and Metastatic Cancer and In-Hospital Outcomes

Figure 3 details the results of the multivariable logistic regressions assessing the associations between non-metastatic and metastatic cancer and in-hospital outcomes. Non-metastatic cancer was associated with a 23% increase in the odds of in-hospital mortality (odds ratio (95% confidence interval)—1.23 (1.07–1.42)). This association exhibited significant interactions with IVT and ET: non-metastatic cancer was not associated with increased in-hospital mortality amongst AIS hospitalisations undergoing either IVT or ET. Metastatic cancer was associated with a 2-fold increase in the odds of in-hospital mortality: 2.16 (1.90–2.45). This association also exhibited a significant interaction with ET, but not IVT: metastatic cancer was not associated with increased in-hospital mortality amongst AIS patients undergoing ET. Non-metastatic and metastatic cancers were also associated with increased odds of prolonged hospitalisation and decreased odds of routine discharge.

### 3.4. Association between Non-Metastatic and Metastatic Cancer and In-Hospital Outcomes amongst Patients Undergoing Endovascular Thrombectomy

Figure 4 details the results of the multivariable logistic regression assessing the relationship between non-metastatic and metastatic cancer and in-hospital outcomes amongst AIS patients undergoing ET. In this group, neither non-metastatic nor metastatic cancer was associated with significantly increased odds of in-hospital mortality or prolonged hospitalisation. These relationships did not significantly differ amongst patients receiving pre-treatment with IVT. Similarly, neither non-metastatic nor metastatic cancer was associated with decreased odds of routine home discharge amongst AIS patients undergoing ET. While the relationship between non-metastatic disease and routine home discharge did not differ based on IVT pre-treatment, patients with metastatic disease undergoing IVT before ET were significantly more likely to be discharged home than those only undergoing ET (*p* = 0.031).

### 3.5. Association between the Five Most Common Primary Cancer Types and In-Hospital Outcomes

Figure 5 details the results of the multivariable logistic regression assessing the association between the five most common primary cancer types and in-hospital outcomes, stratifying by metastatic disease status. Respiratory cancers (both non-metastatic—1.88 (1.48–2.40) and metastatic—2.40 (1.90–3.02)), pancreatic cancers (both non-metastatic (1.96 (1.04–3.71)) and metastatic—2.33 (1.61–3.37)) and metastatic colorectal cancer (2.08 (1.21–3.58)) were associated with significantly increased in-hospital mortality. There were no associations between metastatic prostate cancer, breast cancer (both non-metastatic and metastatic) and non-metastatic colorectal cancer and in-hospital mortality. Non-metastatic prostate cancer was associated with decreased odds of in-hospital mortality (0.62 (0.40–0.96)). Respiratory cancers, metastatic pancreatic and metastatic colorectal cancer were associated with increased odds of prolonged hospitalisation. Respiratory cancers, metastatic prostate cancer, pancreatic cancer (both non-metastatic and metastatic) and metastatic colorectal cancer were associated with decreased odds of routine discharge.

## 4. Discussion

In this study, including a sample representative of 1,106,045 acute ischemic stroke admissions between 2015–2017, we have determined the distribution of prevalent cancers as well as their association with in-hospital outcomes. We have also determined how these associations are influenced by the use of revascularisation therapies.

The five most common cancer types were: respiratory (24.42%), prostate (12.77%), breast (8.69%), pancreatic (6.79%) and colorectal cancers (6.41%). We report significant differences in stroke treatment according to cancer diagnosis, with patients with non-metastatic cancer at 23% lower odds of receiving IVT and 20% lower odds of receiving ET. The disparities in treatments were even greater in patients with metastatic cancer who were at 61% lower odds of receiving IVT compared to patients without cancer. Patients with cancer were at increased risk of adverse AIS outcomes. Non-metastatic cancer was associated with 23% increased odds of in-hospital mortality, which was mainly driven by pancreatic (96% increased odds) and respiratory (88% increased odds) cancers. This association was entirely offset by both IVT and ET. Metastatic cancer was associated with 2-fold increased odds of in-hospital mortality, which was mainly driven by respiratory (2.43-fold increase), pancreatic (2.37-fold increase) and colorectal (2.11-fold) cancers. This association was only offset by ET. Non-metastatic cancer was also associated 16% decreased odds of routine discharge, which was mainly driven by pancreatic (48% decreased odds) and respiratory (35%) cancers. This association was offset by IVT but not by ET. Metastatic cancer was associated with 40% decreased odds of routine discharge, mainly driven by pancreatic (54% decreased odds), respiratory (47%), colorectal (32%), prostate (28%) and breast cancers (27%). This association was offset by ET and ET + IVT, but not by IVT alone. Similarly, non-metastatic and metastatic cancers were also associated with prolonged hospitalisation.

The association between co-morbid cancer and AIS outcomes has been previously evaluated on smaller, single-centre cohorts [4,31,32]. A previous investigation including ~5000 AIS patients out of whom 1.46% had co-morbid cancer found a 3.7-fold increase in the odds of in-hospital mortality [4]. Another study including 468 AIS patients with co-morbid cancer found that metastatic disease was independently associated with a 4.5-fold increase in the risk of 6-month mortality compared to non-metastatic cancers [32]. Furthermore, gastric and pancreatic cancers were associated with increased mortality risk compared to other cancer types [32]. Our results complement these previous findings by providing a comprehensive description of the association between these disease entities based on a large, national real-world sample of AIS patients and highlight disparities in provision of evidence-based therapies.

Furthermore, our results provide additional insight into these relationships by exploring differences based on the presence of metastases and between different primary types. We found 23% increased odds of in-hospital mortality and 16% decreased odds of routine home discharge associated with non-metastatic cancer, which may be attributed to a higher proportion of cryptogenic strokes [4] with worse prognosis [33] and increased hypercoagulability leading to complications such as venous thromboembolism, recurrent stroke [34] and a greater risk of haemorrhagic transformation. Another important contributor to the increased risk of adverse outcomes was significant differences in the receipt of IVT and ET, with patients with cancer consistently less likely to receive revascularisation therapies. This may be related to differences in the time to hospital presentation between patients with and without cancer, as well as likely increased prevalence of contraindications to IVT, such as thrombocytopaenia, previous anticoagulation and recent surgery, amongst cancer patients.

Previous large clinical trials assessing the use of IVT for AIS revascularisation provide no specific information regarding AIS patients with cancer [9,10,11,12]. Nevertheless, several observational studies including AIS patients with co-morbid cancer have found no association between IVT and in-hospital mortality, major bleeding or functional outcomes [13,14,35,36,37,38], while current guidelines recommend IVT in patients with systemic malignancy provided they have a life expectancy >6 months and no other contraindications [8]. We found that even after comprehensive adjustment, patients with co-morbid cancer were less likely to receive IVT, suggesting that the lower rates of IVT in this population may not be fully explained by a higher prevalence of contraindications. This may reflect the fact that treating clinicians may be hesitant to use IVT solely based on cancer status, even in those with non-metastatic disease.

Our work shows for the first time that IVT offset the increased in-hospital mortality and decreased odds of routine discharge associated with non-metastatic cancer, providing supportive evidence for IVT therapy in these patients. Nevertheless, the associations between metastatic cancer and adverse in-hospital outcomes were not offset by IVT, suggesting that AIS patients with metastatic cancer may not fully benefit from this therapy. This may relate to the fact that strokes in this patient group may be misdiagnosed cerebral metastases presenting as a stroke mimic. Furthermore, a greater likelihood of haemorrhagic transformation in this patient group may offset any benefit with thrombolysis. However, we also found that IVT pre-treatment was associated with a significant increase in the odds of routine home discharge in patients with metastatic cancer undergoing ET, suggesting a potentially different treatment effect in this patient group. Nevertheless, these results need to be interpreted in the light of the limitations inherent to the small sample size of the included patient group with cancer undergoing ET.

As with IVT, clinical trials assessing the use of ET in AIS provide no specific data regarding its use in cancer patients [7]. Nevertheless, a few small retrospective observational studies have assessed the use of ET in Asian AIS patients with cancer, reaching different conclusions [16,17,32]. While similar acute outcomes were found amongst AIS patients with and without co-morbid cancer undergoing ET in two studies [16,17], a third study found significantly worse functional outcomes in cancer patients treated with ET [18]. Furthermore, it has been postulated that AIS patients with cancer may have different clot composition, which may hinder successful recanalisation in this population [16].

Our analysis shows that patients with non-metastatic cancer were significantly less likely to receive ET, while those with metastatic disease were equally likely to receive ET compared to patients without cancer. These differences may reflect the fact that metastatic cancer patients may be more likely to present with cardioembolic strokes caused large artery occlusion, rendering them more likely candidates for ET. Nevertheless, in the absence of more granular stroke syndrome data, we could not assess this hypothesis and further research is thus warranted. Our study provides for the first time an analysis of the relationship between cancer and ET in AIS patients using a large, real-world and contemporary sample. ET offset the excess odds of in-hospital adverse outcomes associated both with non-metastatic and metastatic cancers, suggesting that ET may be a successful strategy in eligible patients with cancer, especially in those with metastases who may not fully benefit from IVT. Our findings also suggest that concomitant IVT and ET treatment further improves the odds of routine home discharge in patients with metastatic cancer and may be considered in such patients without specific contraindications to either revascularisation strategy.

Our study has several strengths, such as including a large sample representative of >1 million AIS admissions between late 2015–2017 across the United States as well as having adjusted for a wide range of important confounders. Our results thus reflect contemporary stroke management, including the more widespread adoption of ET and thus allow the generalisation of clinical implications to patients with similar characteristics. Our findings show that both non-metastatic and metastatic disease are associated with significant increases in in-hospital mortality, prolonged hospitalisation and decreased odds of routine home discharge. This highlights that cancer patients warrant thorough primary prevention, since they are not only more likely to suffer an incident stroke, but also at higher odds of acute adverse stroke outcomes. Given our large sample size, our study is able to provide more granular information regarding individual associations between each primary cancer type and adverse acute outcomes.

Our study highlights inequalities in the receipt of evidence-based reperfusion therapies in cancer patients. Such differences have also been described in other cardiovascular conditions such as myocardial infarction [28]. We report that cancer patients offered treatment with IVT or ET may derive a benefit and that IVT may offset the non-metastatic cancer-associated excess odds of adverse outcomes. Furthermore, ET offsets the excess odds associated with both non-metastatic and metastatic disease. Along with previous findings, our study also suggests that co-morbid cancer should not represent a contraindication to AIS revascularisation therapies in itself.

Naturally, our study also has limitations. Having used administrative data, we defined AIS using ICD-10 codes and thus lacked more detailed information regarding stroke severity or classification. Nevertheless, all analyses were adjusted for a wide range of confounders including some important predictors of severe or cardioembolic stroke, such as atrial fibrillation and heart failure [39,40], which may have partly accounted for stroke severity or classification. Furthermore, we also lacked more information regarding cancer staging except for the presence of metastases. We were thus unable to further stratify our analyses by cancer stage. Our database also did not capture treatments such as antithrombotic therapy, which may contribute to the differences in outcomes. Finally, our study only assessed in-hospital outcomes and further research including is also required to characterise the long-term stroke outcomes after hospital discharge associated with co-morbid cancer as well as their interaction with revascularisation strategies.

## 5. Conclusions

In conclusion, in this study including a sample representative of 1.1 million AIS admissions across the United States between 2015–2017, we reported that patients with cancer represent one in thirty acute stroke admissions in the United States and are associated with an increased risk of mortality. We also report that, even after adjustment for differences in comorbidity, patients with cancer are less likely to be offered revascularisation therapies. These disparities in care may contribute to some of the observed adverse outcomes associated with a cancer diagnosis. Nevertheless, IVT offset the non-metastatic cancer-associated excess odds of mortality, while ET offset both the non-metastatic and metastatic cancer-associated excess odds of mortality. Both non-metastatic and metastatic cancers were associated with increased odds of prolonged hospitalisation and decreased odds of routine discharge. IVT and ET are useful strategies to improve in-hospital outcomes in this population and should be offered routinely in cancer patients unless otherwise contraindicated.

## Figures and Tables

**Figure 1 cancers-13-02193-f001:**
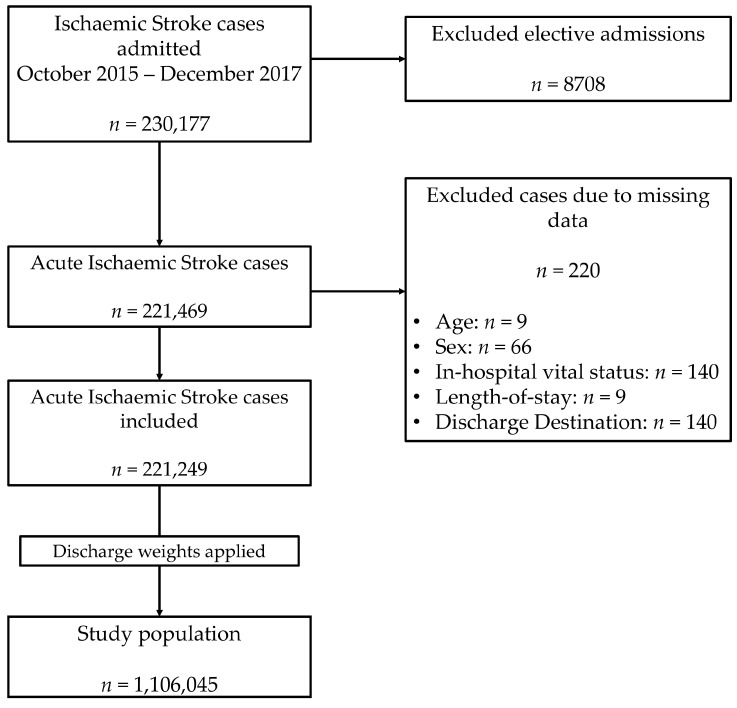
Patient population flowchart.

**Figure 2 cancers-13-02193-f002:**
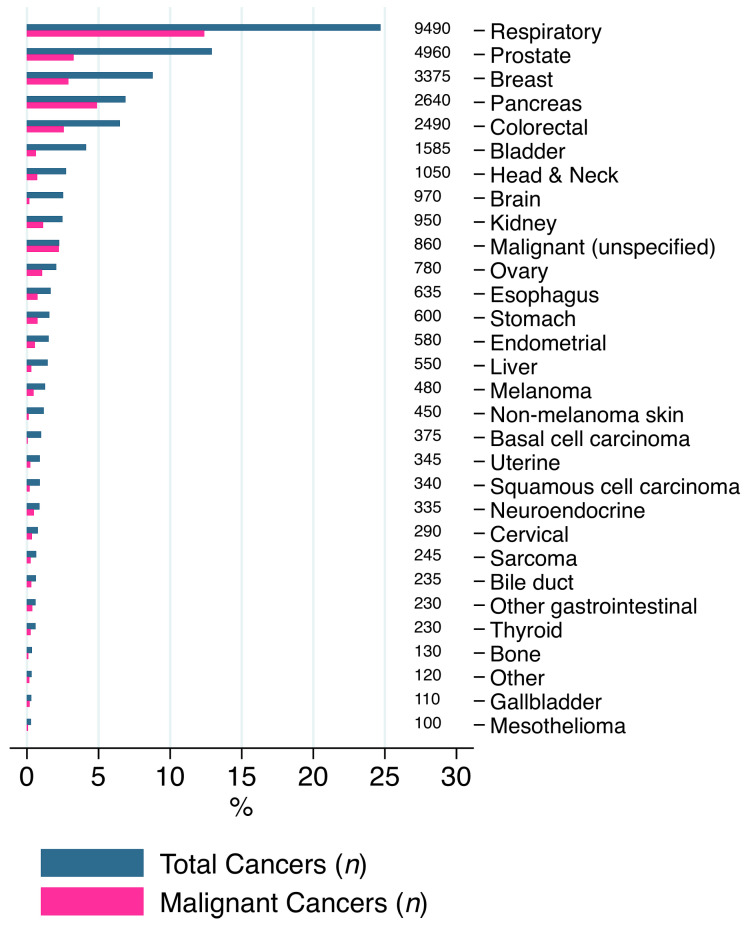
Distribution of each primary cancer type amongst the included sample of acute ischaemic stroke patients, representative of 1,106,045 patients.

**Figure 3 cancers-13-02193-f003:**
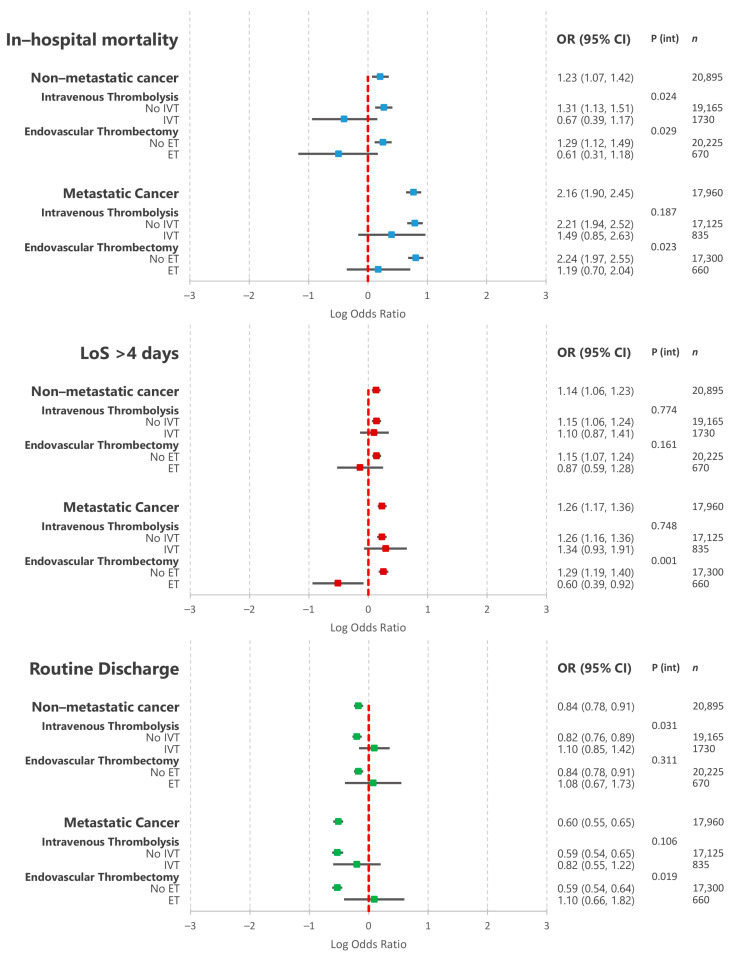
Results of multivariable logistic regressions assessing the association between co-morbid cancer (non-metastatic and metastatic) and acute ischaemic stroke in-hospital outcomes as well as the interaction with revascularisation therapies. OR—odds ratio; CI—confidence interval; IVT—intravenous thrombolysis; ET—endovascular thrombectomy; P (int)—*p* value for interaction term.

**Figure 4 cancers-13-02193-f004:**
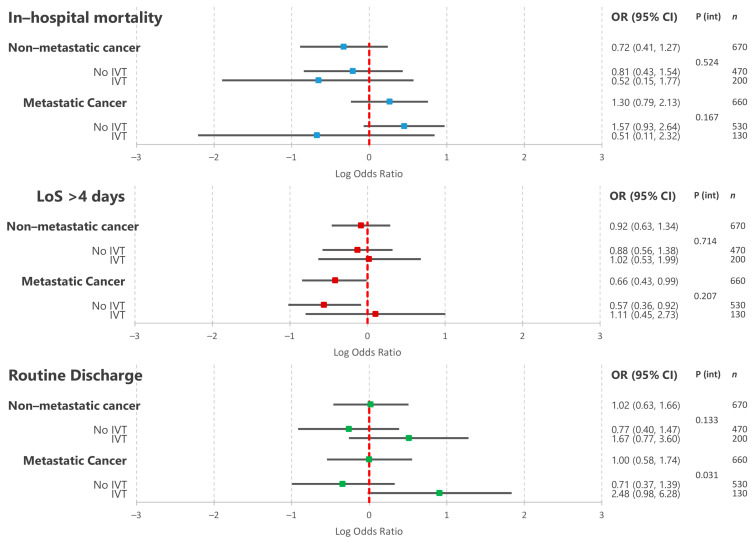
Results of multivariable logistic regressions assessing the association between co-morbid cancer (non-metastatic and metastatic) and acute ischaemic stroke in-hospital outcomes amongst AIS patients undergoing endovascular thrombectomy. Separate models containing interaction terms with intravenous thrombolysis therapy were constructed to determine whether outcomes were modified by IVT pre-treatment. OR—odds ratio; CI—confidence interval; IVT—intravenous thrombolysis; ET—endovascular thrombectomy; P (int)—*p* value for interaction term.

**Figure 5 cancers-13-02193-f005:**
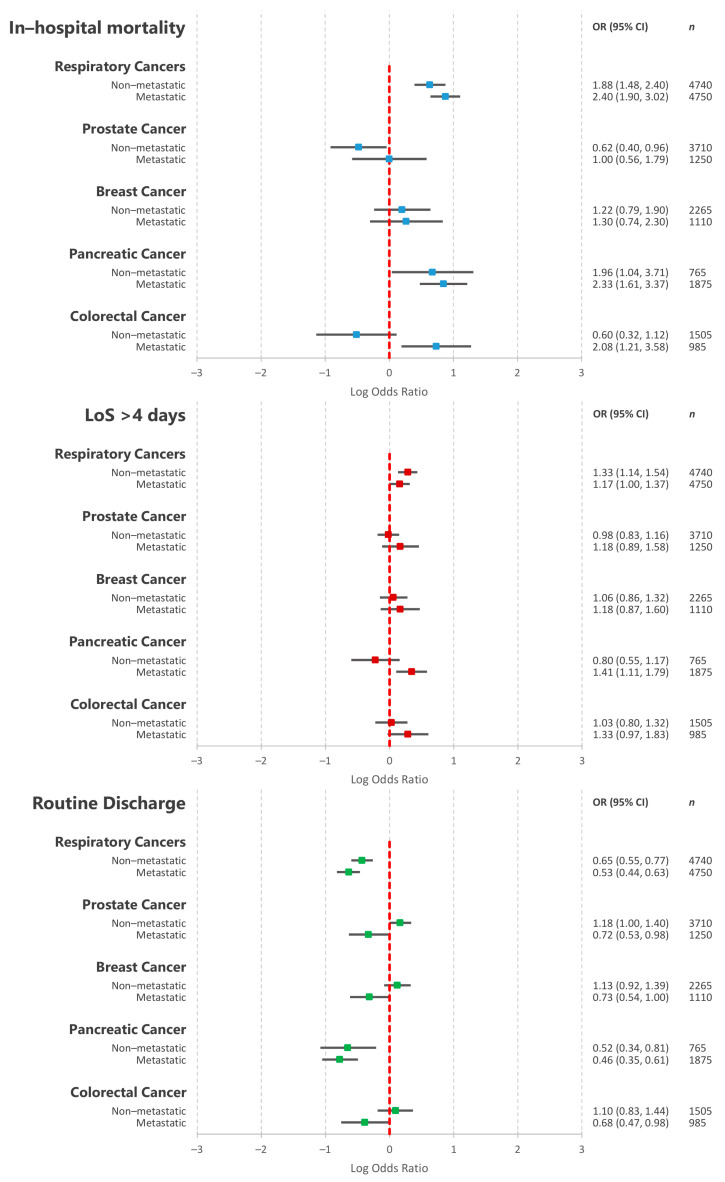
Results of multivariable logistic regressions assessing the association between the 5 most commonly occurring cancer types in the included cohort and acute ischaemic stroke in-hospital mortality as well as the interaction with revascularisation therapies. OR—odds ratio; CI—confidence interval; IVT—intravenous thrombolysis; ET—endovascular thrombectomy.

**Table 1 cancers-13-02193-t001:** Patient characteristics on admission, stratified by either co-existent non-metastatic or metastatic cancer. Further descriptive statistics are detailed in Appendix A. Statistically significant *p* values (<0.05) are highlighted in **bold**.

Variation	All	No Active Cancer	Non-Metastatic Cancer	Metastatic Cancer	*p* Value
*n*	1,106,045	1,067,190	20,895	17,960	
Age (years), median (IQR)	72 (61–82)	71 (60–82)	75 (67–83)	70 (62–78)	**<0.001**
Length-of-stay (days), median (IQR)	3 (2–6)	3 (2–6)	4 (2–7)	4 (2–7)	**<0.001**
Sex Female, *n* (%)	557,595 (50.41)	538,635 (50.47)	9795 (46.88)	9165 (51.03)	**<0.001**
**Ethnicity**					**<0.001**
White	735,330 (66.48)	707,550 (66.30)	15,005 (71.81)	12,775 (71.13)	
Black	183,090 (16.55)	177,790 (16.66)	2880 (13.78)	2420 (13.47)	
Hispanic	84,950 (7.68)	82,700 (7.75)	1240 (5.93)	1010 (5.62)	
Asian or Pacific Islander	31,635 (2.86)	30,500 (2.86)	530 (2.54)	605 (3.37)	
Native American	4700 (0.42)	4570 (0.43)	60 (0.29)	70 (0.39)	
Other	27,460 (2.48)	26,535 (2.49)	410 (1.96)	515 (2.87)	
**ELIXHAUSER CO-MORBIDITIES**, *n* (%)					
Congestive Heart Failure	172,170 (15.57)	166,950 (15.64)	3225 (15.43)	1995 (11.11)	**<0.001**
Valvular Disease	110,540 (9.99)	106,640 (9.99)	2300 (11.01)	1600 (8.91)	**0.008**
Pulmonary Circulation Disease	8460 (0.76)	6710 (0.63)	555 (2.66)	1195 (6.65)	**<0.001**
Peripheral Vascular Disease	112,065 (10.13)	107,955 (10.12)	2470 (11.82)	1640 (9.13)	**<0.001**
Paralysis	112,895 (10.21)	108,920 (10.21)	2260 (10.82)	1715 (9.55)	0.178
Other Neurological Disorders	6620 (0.60)	6275 (0.59)	210 (1.01)	135 (0.75)	**0.001**
Chronic Pulmonary Disease	174,180 (15.75)	165,690 (15.53)	4795 (22.95)	3695 (20.57)	**<0.001**
Diabetes (without chronic complications)	210,220 (19.01)	203,765 (19.09)	3520 (16.85)	2935 (16.34)	**<0.001**
Diabetes (with chronic complications)	214,400 (19.38)	209,135 (19.60)	3095 (14.81)	2170 (12.08)	**<0.001**
Hypothyroidism	159,160 (14.39)	153,810 (14.41)	2950 (14.12)	2400 (13.36)	0.193
Renal Failure	181,950 (16.45)	175,960 (16.49)	3670 (17.56)	2320 (12.92)	**<0.001**
Liver Disease	18,310 (1.66)	17,275 (1.62)	550 (2.63)	485 (2.70)	**<0.001**
Peptic Ulcer Disease	7695 (0.70)	7400 (0.69)	175 (0.84)	120 (0.67)	0.527
Acquired Immune Deficiency Syndrome	2395 (0.22)	2280 (0.21)	100 (0.48)	15 (0.08)	**<0.001**
Lymphoma	5315 (0.48)	4915 (0.46)	220 (1.05)	180 (1.00)	**<0.001**
Rheumatoid Arthritis/Collagen Vascular Disease	30,150 (2.73)	29,240 (2.74)	520 (2.49)	390 (2.17)	0.075
Coagulopathy	41,405 (3.74)	37,105 (3.48)	1560 (7.47)	2740 (15.26)	**<0.001**
Obesity	145,465 (13.15)	142,575 (13.36)	1775 (8.49)	1115 (6.21)	**<0.001**
Weight loss	44,030 (3.98)	39,685 (3.72)	1795 (8.59)	2550 (14.20)	**<0.001**
Fluid and electrolyte disorders	246,680 (22.30)	235,750 (22.09)	5200 (24.89)	5730 (31.90)	**<0.001**
Anaemia (chronic blood loss)	4025 (0.36)	3590 (0.34)	240 (1.15)	195 (1.09)	**<0.001**
Anaemia (deficiency)	133,005 (12.03)	123,720 (11.59)	4375 (20.94)	4910 (27.34)	**<0.001**
Alcohol abuse	49,375 (4.46)	48,080 (4.51)	800 (3.83)	495 (2.76)	**<0.001**
Drug abuse	28,985 (2.62)	28,365 (2.66)	355 (1.70)	265 (1.48)	**<0.001**
Psychoses	26,255 (2.37)	25,490 (2.39)	440 (2.11)	325 (1.81)	**0.041**
Depression	124,635 (11.27)	120,280 (11.27)	2330 (11.15)	2025 (11.28)	0.971
Hypertension	946,140 (85.54)	916,430 (85.87)	16,975 (81.24)	12,735 (70.91)	**<0.001**
**PROCEDURES**, *n* (%)					
Thrombectomy	34,420 (3.11)	33,090 (3.10)	670 (3.21)	660 (3.67)	0.139
Thrombolysis	103,600 (9.37)	101,035 (9.47)	1730 (8.28)	835 (4.65)	**<0.001**
**OUTCOMES**, *n* (%)					
In-hospital mortality	43,545 (3.94)	40,545 (3.80)	1230 (5.89)	1770 (9.86)	**<0.001**
Length-of-stay >4 days	380,605 (34.41)	363,430 (34.05)	8680 (41.54)	8495 (47.30)	**<0.001**
Routine Discharge	394,105 (35.99)	384,490 (36.39)	5600 (26.94)	4015 (22.52)	**<0.001**

## Data Availability

The data that support the findings of this study are available from the corresponding author upon reasonable request.

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
