# Peer review of "Prognosis of Acute Ischaemic Stroke Patients with Cancer: A National Inpatient Sample Study"

_cancers, 2021, doi:10.3390/cancers13092193_

Round 1

Reviewer 1 Report

The topic is interesting, useful and the study is designed well. The manuscript is written correctly. There are no conflicts of interest. The methodology and statistical methods are described accurately and clearly. The advantage of this study is the analysis of a large group of patients. These results are very useful in clinical practice. I have no critical comments.

Author Response

Thank you for taking the time to review our manuscript. We are pleased you have found our work of interest and clinical importance. We would also like to thank you for your positive comments.

Reviewer 2 Report

This article discusses the prognosis of AIS patients with cancer and the effects of intravenous thrombolysis (IVT) and endovascular thrombectomy (ET).

The paper analyzes mortality, length of hospital stay, and return home rates, but focuses primarily on mortality. The mortality rate is of course important to evaluate, but the return home rate is just as important. I would like the authors to evaluate the results of their analysis to see how effective and influential IVT and ET were.

We also believe that ET can be divided into two groups: those who received IVT and those who did not. We would like to see an analysis of how the results of ET preceded by IVT affected mortality, length of hospital stay, and return home rates.

Author Response

We have highlighted the changes in the text, where appropriate, as follows: (additions in blue, deletions in red):.

Reviewer 2

This article discusses the prognosis of AIS patients with cancer and the effects of intravenous thrombolysis (IVT) and endovascular thrombectomy (ET).

The paper analyzes mortality, length of hospital stay, and return home rates, but focuses primarily on mortality. The mortality rate is of course important to evaluate, but the return home rate is just as important. I would like the authors to evaluate the results of their analysis to see how effective and influential IVT and ET were.

Response: Thank you for taking the time to review our work. We would also like to thank you for your comments and providing insightful and useful feedback which we have addressed in this resubmission. We agree with the reviewer that return home rates, quantified by routine home discharges in our study, is an equally important outcome to discuss as in-hospital mortality. We have therefore revised our discussion section, giving equal emphasis to both outcomes.

Under DISCUSSION, page 13, lines 291-311:

“The five most common cancer types were: respiratory (24.42%), prostate (12.77%), breast (8.69%), pancreatic (6.79%) and colorectal cancers (6.41%). We report significant differences in stroke treatment according to cancer diagnosis, with patients with non-metastatic cancer at 23% lower odds of receiving IVT and 20% lower odds of receiving ET. The disparities in treatments were even greater in patients with metastatic cancer who were at 61% lower odds of receiving IVT compared to patients without cancer. Patients with cancer were at increased risk of adverse AIS outcomes. Non-metastatic cancer was associated with 23% increased odds of in-hospital mortality, which was mainly driven by pancreatic (96% increased odds) and respiratory (88% increased odds) cancers. This association was entirely offset by both IVT and ET. Metastatic cancer was associated with 2-fold increased odds of in-hospital mortality, which was mainly driven by respiratory (2.43-fold increase), pancreatic (2.37-fold increase) and colorectal (2.11-fold) cancers. This association was only offset by ET. Non-metastatic and metastatic cancers were was also associated with increased odds of prolonged hospitalisation and  16% decreased odds of routine discharge, which was mainly driven by pancreatic (48% decreased odds) and respiratory (35%) cancers. This association was offset by IVT but not by ET. Metastatic cancer was associated with 40% decreased odds of routine discharge, mainly driven by pancreatic (54% decreased odds), respiratory (47%), colorectal (32%), prostate (28%) and breast cancers (27%). This association was offset by ET and ET+IVT, but not by IVT alone. Similarly, non-metastatic and metastatic cancers were also associated with prolonged hospitalisation.

Under DISCUSSION, page 13, lines 323-329:

“Furthermore, our results provide additional insight into these relationships by exploring differences based on the presence of metastases and between different primary types. We found 23% increased odds of in-hospital mortality and 16% decreased odds of routine home discharge associated with non-metastatic cancer, which may be attributed to a higher proportion of cryptogenic strokes4 with worse prognosis33  and increased hypercoagulability leading to complications such as venous thromboembolism, recurrent stroke 34 and a greater risk of haemorrhagic transformation.”

Under DISCUSSION, page 13, lines 347-359:

“Our work shows for the first time that IVT offset the increased in-hospital mortality and decreased odds of routine discharge associated with non-metastatic cancer, providing supportive evidence for IVT therapy in these patients. Nevertheless, the associations between metastatic cancer and adverse in-hospital outcomes were not offset by IVT, suggesting that AIS patients with metastatic cancer may not fully benefit from this therapy. This may relate to the fact that strokes in this patient group may be misdiagnosed as cerebral metastases, which may present as a stroke mimic.   Furthermore, a greater likelihood of haemorrhagic transformation in this patient groups may offset any benefit with thrombolysis. However, we also found that IVT pre-treatment was associated with a significant increase in the odds of routine home discharge in patients with metastatic cancer undergoing ET, suggesting a potentially different treatment effect in this patient group. Nevertheless, these results need to be interpreted in the light of the limitations inherent to the small sample size of the included patient group with cancer undergoing ET.

We also believe that ET can be divided into two groups: those who received IVT and those who did not. We would like to see an analysis of how the results of ET preceded by IVT affected mortality, length of hospital stay, and return home rates.

Response: Thank you for highlighting this important point. We agree that pre-treatment with IVT would be an important factor to consider when characterising the relationship between ET and AIS outcomes in cancer patients. We have subsequently performed a further analysis including only AIS patients undergoing ET. We also performed separate models including an interaction term with IVT in order to determine how outcomes were modified IVT pre-treatment. Nevertheless, it is important to highlight this analysis may be limited by the relatively small number of AIS patients with cancer who underwent either ET or ET+IVT.

Under METHODS, page 4, lines 159-166:

2.1.6. Association between Non-Metastatic and Metastatic Cancer and In-Hospital Outcomes amongst patients undergoing Endovascular Thrombectomy

            Further multivariable logistic regressions were performed in order to explore the relationship between co-morbid non-metastatic and metastatic cancer and in-hospital outcomes amongst AIS patients undergoing ET. Separate models including an interaction term with IVT were also computed in order to determine how outcomes were modified by whether these patients also received IVT pre-treatment. All models were adjusted for the covariates listed above.

Under RESULTS, page 10, lines 245-257:

3.4. Association between Non-Metastatic and Metastatic Cancer and In-Hospital Outcomes amongst patients undergoing Endovascular Thrombectomy

            Figure 4 details the results of the multivariable logistic regression assessing the relationship between non-metastatic and metastatic cancer and in-hospital outcomes amongst AIS patients undergoing ET. In this group, neither non-metastatic nor metastatic cancer was associated with significantly increased odds of in-hospital mortality or prolonged hospitalisation. These relationships did not significantly differ amongst patients receiving pre-treatment with IVT. Similarly, neither non-metastatic nor metastatic cancer was associated with decreased odds of routine home discharge amongst AIS patients undergoing ET. While the relationship between non-metastatic disease and routine home discharge did not differ based on IVT pre-treatment, patients with metastatic disease undergoing IVT before ET were significantly more likely to be discharged home than those only undergoing ET (P = 0.031).

Under DISCUSSION, page 14, lines 347-359:

“Our work shows for the first time that IVT offset the increased in-hospital mortality and decreased odds of routine discharge associated with non-metastatic cancer, providing supportive evidence for IVT therapy in these patients. Nevertheless, the associations between metastatic cancer and adverse in-hospital outcomes were not offset by IVT, suggesting that AIS patients with metastatic cancer may not fully benefit from this therapy. This may relate to the fact that strokes in this patient group may be misdiagnosed as cerebral metastases, which may present as a stroke mimic.   Furthermore, a greater likelihood of haemorrhagic transformation in this patient groups may offset any benefit with thrombolysis. However, we also found that IVT pre-treatment was associated with a significant increase in the odds of routine home discharge in patients with metastatic cancer undergoing ET, suggesting a potentially different treatment effect in this patient group. Nevertheless, these results need to be interpreted in the light of the limitations inherent to the small sample size of the included patient group with cancer undergoing ET.

Under DISCUSSION, page 14, lines 368-382:

“Our analysis shows that patients with non-metastatic cancer were significantly less likely to receive ET, while those with metastatic disease were equally likely to receive ET compared to patients without cancer. These differences may reflect the fact that metastatic cancer patients may be more likely to present with cardioembolic strokes caused large artery occlusion, rendering them more likely candidates for ET. Nevertheless, in the absence of more granular stroke syndrome data, we could not assess this hypothesis and further research is thus warranted. Our study provides for the first time an analysis of the relationship between cancer and ET in AIS patients using a large, real-world and contemporary sample. ET offset the excess odds of in-hospital adverse outcomes associated both with non-metastatic and metastatic cancers, suggesting that ET may be a successful strategy in eligible patients with cancer, especially in those with metastases who may not fully benefit from IVT. Our findings also suggest that concomitant IVT and ET treatment further improves the odds of routine home discharge in patients with metastatic cancer and may be considered in such patients without specific contraindications to either revascularisation strategy.
